# Accuracy of Noninvasive Diagnostic Tests for the Detection of Significant and Advanced Fibrosis Stages in Nonalcoholic Fatty Liver Disease: A Systematic Literature Review of the US Studies

**DOI:** 10.3390/diagnostics12112608

**Published:** 2022-10-27

**Authors:** Dhaval Gosalia, Vlad Ratziu, Filip Stanicic, Djurdja Vukicevic, Vladimir Zah, Nadege Gunn, Dina Halegoua-DeMarzio, Tram Tran

**Affiliations:** 1Department of Commercial Strategy and Operations, Glympse Bio, Cambridge, MA 02140, USA; 2Department of Medicine, Medicine Sorbonne University, 75013 Paris, France; 3Health Economics and Outcomes Research Department, ZRx Outcomes Research Inc., Mississauga, ON L5A 2X7, Canada; 4Department of Hepatology, Impact Research Institute, Waco, TX 76710, USA; 5Department of Medicine, Sidney Kimmel Medical College, Thomas Jefferson University, Philadelphia, PA 19107, USA; 6Department of Medicine, UCLA Santa Monica Medical Center, Santa Monica, CA 90404, USA; 7Department of Medicine, UCLA David Geffen School of Medicine, Los Angeles, CA 90095, USA

**Keywords:** noninvasive diagnostic tools, sensitivity, specificity, liver fibrosis, NAFLD

## Abstract

Background: The purpose of this systematic literature review (SLR) was to evaluate the accuracy of noninvasive diagnostic tools in detecting significant or advanced (F2/F3) fibrosis among patients with nonalcoholic fatty liver (NAFL) in the US healthcare context. Methods: The SLR was conducted in PubMed and Web of Science, with an additional hand search of public domains and citations, in line with the PRISMA statement. The study included US-based original research on diagnostic test sensitivity, specificity and accuracy. Results: Twenty studies were included in qualitative evidence synthesis. Imaging techniques with the highest diagnostic accuracy in F2/F3 detection and differentiation were magnetic resonance elastography and vibration-controlled transient elastography. The most promising standard blood biomarkers were NAFLD fibrosis score and FIB-4. The novel diagnostic tools showed good overall accuracy, particularly a score composed of body mass index, GGT, 25-OH-vitamin D, and platelet count. The novel approaches in liver fibrosis detection successfully combine imaging techniques and blood biomarkers. Conclusions: While noninvasive techniques could overcome some limitations of liver biopsy, a tool that would provide a sufficiently sensitive and reliable estimate of changes in fibrosis development and regression is still missing.

## 1. Introduction

Nonalcoholic fatty liver disease (NAFLD) is the most common chronic liver disease worldwide, affecting around a quarter of the general population and a third of the United States (US) population [1,2,3].

Over time, NAFLD may progress to nonalcoholic steatohepatitis (NASH), which is considered a more progressive form of the disease. NASH is histologically defined as hepatic steatosis, inflammation, and ballooning (enlarged cells with rarefied cytoplasm [4]) with or without fibrosis, caused by lipotoxicity of accumulated lipids in hepatocytes and immune cell activation [5,6]. It is estimated that NASH affects up to 6.5% of the general population worldwide and 3–4% of the US population [2,7]. The diagnosis is more common among obese and diabetic patients, occurring in around 30% and 65% of cases, respectively [8].

One of the most common complications of NAFLD and NASH is liver fibrosis, occurring in more than a third of NASH patients over a 5-year period [9]. The level of liver fibrosis in NAFLD is commonly scored using the NASH CRN system, where fibrosis stage 0 represents no fibrosis; stage 1 demarks pericellular fibrosis; stage 2 denotes centrilobular and periportal fibrosis; stage 3 is bridging fibrosis, and stage 4 represents cirrhosis [10]. Fibrosis stage F2 or higher (F2+) is considered significant fibrosis. Advanced fibrosis traditionally refers to stage F3 or higher (F3+) [11].

About 8% of the general population and 13% of the high-risk population are assumed to have undetected advanced fibrosis [12,13,14]. A recent population analysis of data from National Health and Nutrition Examination Survey (NHANES) established that around 7.5% of NAFLD patients had advanced fibrosis [15]. Liver fibrosis is the most important prognostic factor in the course of NAFLD, as it is the only pathological finding that correlates with hepatic decompensation events and liver-related mortality [16,17]. Early and accurate diagnosis and staging of fibrosis in NAFLD and NASH patients, particularly those with significant and advanced fibrosis (F2 and F3 stages), is necessary to determine the patient’s prognosis and guide clinical decision-making [11,18].

Focusing on universal access to a consistently accurate and minimally invasive diagnosis of patients with significant or advanced fibrosis would ensure appropriate disease management and a better prognosis. Studies summarizing evidence regarding noninvasive diagnostic tests’ ability to accurately detect F2+ and F3+ fibrosis stages in NAFLD and NASH patients are lacking within the US-based studies.

The objective of the current systematic literature review (SLR) was to collect, summarize, and interpret published evidence from US studies on the accuracy of currently available diagnostic tests in detecting and longitudinally monitoring F2+/F3+ fibrosis stages in NAFLD and NASH patients. As the prevalence and natural course of NAFLD vary across the continents, races, and ethnic groups, the study focused on original research articles conducted in the US, assuming a similar demographic distribution across the included studies and enhancing the comparability of their results. The availability of different imaging techniques and biomarkers also varies across different countries and the study targeted currently existing noninvasive diagnostic tests available in the US.

## 2. Materials and Methods

### 2.1. Data Sources and Selection Criteria

The key literature databases for the SLR were the Medical Literature Analysis and Retrieval System Online (MEDLINE^®^), assessed via PubMed and Web of Science (January 2016 through May 2022). As all the outcomes were considered time-sensitive and sensitive to the evolving methodological approaches, a 6-year time constraint was applied to provide a current update of existing literature reviews. In addition, a hand search was performed across publicly available domains (e.g., Google Scholar) and reference lists to ensure all relevant studies were included. Only primary original research studies were considered, while SLRs, meta-analyses, narrative reviews, and guidelines were excluded. Selection criteria are shown in Table 1.

### 2.2. Search Strategy

The search query (Appendix A) was constructed to address diagnostic tool accuracy in detection and tracking F2+ and F3+ fibrosis stages following population, intervention, comparators, and outcomes (PICO) criteria (Table 2).

The detailed search strategy and yielded hits are presented in the Appendix A. The population included patients with NAFLD and/or NASH, also considering the underlying stages of the disease, i.e., nonalcoholic steatosis, fibrosis, and cirrhosis. All diagnostic tests, imaging techniques, and biomarkers were considered to explore the accuracy of diagnostic tools in terms of sensitivity, specificity, positive and negative predictive values (PPV and NPV), and area under the receiver operating characteristic (AUROC) curve.

### 2.3. Data Review and Extraction

The review was conducted in accordance with the Preferred Reporting Items for Systematic Reviews and Meta-Analysis (PRISMA) guidelines. Two independent reviewers performed the database search, abstract and title search, and full-text screening. A third reviewer resolved any disagreements. Predefined extraction tables were used for data collection and evidence summary.

## 3. Results

The study selection process is shown in the PRISMA flow diagram (Figure 1). After excluding duplicates, 1633 studies were title and abstract screened, 415 publications were full-text screened, and 20 studies were selected for data extraction.

The SLR resulted in 20 original US-based studies that evaluated a wide spectrum of noninvasive imaging modalities and biomarkers for the detection of F2+ and F3+ stage fibrosis. The characteristics of included studies are presented in Table 3 while the full list of diagnostic techniques is presented in Figure 2.

### 3.1. Imaging Techniques

The SLR identified eight studies that explored the diagnostic accuracy and capabilities of imaging techniques in terms of significant (F2+ fibrosis stages) and advanced (F3+ fibrosis stages) fibrosis detection (Table 3). The diagnostic accuracy of imaging techniques in detection of significant and advanced fibrosis is presented in Table 4 and Table 5.

#### 3.1.1. Vibration-Controlled Transient Elastography (VCTE)

In a randomized clinical trial that tested baseline performance of different noninvasive diagnostic techniques, Harrison et al. reported low predictive value of VCTE in differentiating fibrosis stages in NASH patients with biopsy-proven F1–F3 fibrosis (AUROC 0.630 for F2+, 0.650 for F3+) [20]. On the other hand, in a prospective study of liver transplant recipients, Siddiqui et al. found that VCTE detects significant and advanced fibrosis with reliable accuracy. By fixing the sensitivity at 90%, the authors demonstrated its potential to be used as a rule-out tool for significant fibrosis in case of negative results (cutoff 7.4 kPa) among post-liver transplantation patients. It was shown that VCTE with cutoff value of 10.5 kPa was a better ruling-out technique for advanced fibrosis than for significant fibrosis (AUROC 0.940 vs. 0.870, respectively). Still, when the specificity was fixed at 90%, the method yielded a low PPV for both significant and advanced fibrosis (67% and 64%), implying that the tool cannot be reliably used for ruling in the higher-grade fibrosis stages in diagnostic practice [23]. Similarly, a single-center retrospective cohort study by Trowel et al. reported reliable accuracy and rule-out potential of VCTE in advanced fibrosis [25].

#### 3.1.2. Shear Wave Elastography (SWE)

In a retrospective analysis by Ozturk et al., SWE with a cutoff value of 8.4 kPa performed well in detecting significant and advanced fibrosis in patients with suspected or diagnosed NAFLD, concluding that SWE may be useful in detecting the patients at risk of liver morbidity and mortality [22]. Zhang et al. reported better diagnostic performance of SWE in detecting significant and advanced fibrosis among patients with diagnosed or suspected NAFLD. This cross-sectional study pointed out the potential of using SWE as a rule-out diagnostic tool for F2+ and F3+ fibrosis stages (cutoff 1.49 m/s, and 1.46 m/s, respectively). However, using cutoffs with specificity ≥ 90% for significant and advanced fibrosis incorrectly classified approximately every second patient with a positive test result (58.8% and 55.6% PPV, respectively) due to the small prevalence of the condition in the tested sample [26].

#### 3.1.3. Magnetic Resonance Elastography (MRE)

Zhang et al. showed that MRE at a 2.77 kPa cutoff value (for both fibrosis stages) could be used as an ideal rule-out diagnostic tool, while at a fixed specificity of ≥90%, it performed significantly better than SWE in accurately detecting patients with fibrosis F2+ and F3+ stages. Still, the study concluded that neither of the two techniques performed well enough to replace biopsy in detecting significant and advanced fibrosis [26]. In a retrospective analysis of patients with suspected or diagnosed NAFLD from two medical centers, Tang et al. reported almost perfect diagnostic accuracy of MRE in detecting advanced fibrosis (cutoffs of 3.6 kPa and 3.65 kPa). Based on these results, the MRE correctly detected the absence of F3+ fibrosis in nearly all patients, with lower but acceptable rule-in potential [24].

Jayakumar et al. reported higher accuracy of MRE in tracking fibrosis improvement than progression among patients diagnosed with F2 and F3 fibrosis stages at ≥0% reduction cutoff (AUROC 0.790). The tool performed better in detecting improvement among F3 patients than F2 patients, with a high difference in specificity (86% and 33%, respectively) but similar sensitivity (69% and 60%, respectively). The accuracy of detecting F2 and F3 fibrosis progression was modest at ≥0% improvement cutoff (AUROC 0.570), but a high NPV value (88%) implies the method can still be used to rule out the fibrosis progression [21].

#### 3.1.4. Magnetic Resonance Imaging-Derived Liver Surface Nodularity (MRI-Derived LSN) Score

A single-arm prospective study performed by Catania et al. showed that MRI-derived LSN score was a reliable tool in detecting significant and advanced fibrosis among patients with NAFLD (AUROCs 0.800 for F2+, 0.860 for F3+) [19].

Diagnostic accuracy of imaging techniques is presented in Table 4 for significant fibrosis and Table 5 for advanced fibrosis.

### 3.2. Established Fibrosis Scores and Biomarkers

Our SLR identified 11 studies that reported diagnostic accuracy of established general scores and biomarkers in detecting significant and advanced fibrosis (Table 3). Table 6 and Table 7 summarize the diagnostic performance of the established scores and biomarkers in detecting significant and advanced fibrosis.

#### 3.2.1. NAFLD Fibrosis Score (NFS)

In general, the low diagnostic accuracy of NFS was reported in the included studies with AUROC values ranging from 0.600 to 0.640 for the detection of significant fibrosis [20,30]. Harrison et al. reported a relatively low sensitivity (66%) and specificity (52%) of NFS in significant fibrosis detection among NASH patients at a cutoff value of 0.9. Corey et al. reported an 85% specificity rate with a 67% PPV for NFS [30]. As for advanced fibrosis, Harrison et al. reported even lower NFS accuracy with an AUROC of 0.580 [20]. On the other hand, in the studies by Caussy et al. and Marella et al., AUROC values of NFS for advanced fibrosis ranged from 0.810 to 0.840 [29,32]. Bril et al. and Singh et al. reported slightly lower AUROC values (0.640–0.720) among patients with diabetes mellitus (30, 36). Udelsman et al. demonstrated good accuracy of NFS in the advanced fibrosis detection with almost perfect NPV values for both reported cutoffs (98% and 99%, respectively) among patients who underwent liver biopsy before bariatric surgery, indicating it may be a reliable tool for ruling out advanced fibrosis [35], while Balakrishnan et al. reported good prognostic ability of NFS in detection of advanced fibrosis among a predominantly Hispanic population (AUROC of 0.790) [27].

#### 3.2.2. Fibrosis-4 (FIB-4) Index

In Corey et al., the FIB-4 index performed well in detecting significant fibrosis with a high specificity (88%), PPV (76%), and AUROC of 0.700 [30]. Harrison et al. and Nielsen et al. reported good diagnostic accuracy of the FIB-4 index at a cutoff value of 1.3 for detecting significant and advanced fibrosis with AUROC 0.670–0.790 among patients with NASH [20,33]. Other studies established slightly better accuracy of FIB-4 index in the advanced fibrosis detection among NAFLD patients (AUROC of 0.770–0.880) [27,29,32]. Very high specificity reported in the study by Udelsman et al. indicates that FIB-4 would be a reliable noninvasive tool to rule in patients with advanced fibrosis [35]. Similarly, Bril et al. and Singh et al. showed a good diagnostic accuracy of the FIB-4 among patients with diabetes [28,34].

#### 3.2.3. AST to Platelet Ratio Index (APRI)

APRI showed a low diagnostic accuracy for the detection of significant fibrosis with a general AUROC of 0.660 with a cutoff value > 0.42 [33]. In contrast, for the detection of advanced fibrosis, APRI yielded good diagnostic accuracy (AUROC ranged from 0.680–0.860), with high specificity (75–99%), indicating that it may be a reliable tool to rule in patients with advanced fibrosis [27,28,32,34,35].

#### 3.2.4. BARD Score

Balakrishnan et al. reported moderate diagnostic accuracy of the BARD score in detecting advanced fibrosis among predominantly Hispanic NAFLD patients, with a high sensitivity rate, which indicated the BARD score would be reliable for ruling out advanced fibrosis [27].

#### 3.2.5. Enhanced Liver Fibrosis (ELF) Test

In the study by Harrison et al., the ELF test demonstrated satisfying accuracy for detecting significant (cutoff −0.2) and advanced fibrosis (cutoff −0.1) among NASH patients [20]. Younossi et al. reported very high specificity of the ELF test and good reliability in ruling in patients with advanced fibrosis (cutoffs 9.8 and 11.3) among NAFLD patients with biopsy and VCTE as reference tools [36].

#### 3.2.6. FibroTest

The FibroTest has demonstrated modest accuracy in the detection of advanced fibrosis among patients with NAFLD [28]. At cutoff values < 0.3 and >0.7, the FibroTest demonstrated very high specificity, showing its potential for ruling in patients with advanced fibrosis. The main limitation of the FibroTest is the results between 0.3 and 0.7 would be unclassified [28].

#### 3.2.7. Gamma-Glutamyl Transferase (GGT) Levels

Harrison et al. reported low diagnostic accuracy of serum GGT levels in detecting significant and advanced fibrosis among NASH patients [20]. Kulkarni et al. demonstrated slightly better accuracy in detecting significant fibrosis in terms of sensitivity and specificity among NAFLD patients [31], but in general, GGT was marked as a biomarker with low diagnostic accuracy for fibrosis detection.

#### 3.2.8. Aspartate Aminotransferase/Alanine Aminotransferase Ratio (AST/ALT Ratio)

Nielsen et al. assessed the diagnostic accuracy of the AST/ALT ratio for detecting significant and advanced fibrosis among patients with NASH and liver fibrosis [33]. The study denoted very high sensitivity (90%) with a cutoff value > 0.56 and suggested this biomarker could be reliable for ruling out significant fibrosis [33]. In contrast, the same study showed a low diagnostic accuracy of AST/ALT ratio in advanced fibrosis detection with a cutoff value > 0.78 [33], while Singh et al. demonstrated even lower diagnostic accuracy among T2DM patients with NAFLD [34].

#### 3.2.9. AST and ALT Levels

In Harrison et al.’s study, AST and ALT levels demonstrated low accuracy for detecting significant and advanced fibrosis among NASH patients (AUROC ranged from 0.550–0.660) [20]. On the other hand, Bril et al. reported very good accuracy of plasma AST levels in detecting advanced fibrosis among T2DM patients (AUROC of 0.850) at cutoff points of 40 U/L and 38 U/L [28].

Diagnostic accuracy of established fibrosis scores and biomarkers is presented in Table 6 for significant fibrosis and Table 7 for advanced fibrosis.

### 3.3. Novel Biomarkers

The SLR identified eight US studies that evaluated the diagnostic accuracy of novel biomarkers in the detection of significant or advanced fibrosis (Table 3). The diagnostic capabilities of novel biomarkers in the detection of significant and advanced fibrosis are presented in Table 8 and Table 9.

#### 3.3.1. Cytokeratine-18 (CK-18) Fragments M30 and M65

Harrison et al. evaluated the diagnostic accuracy of CK-18 fragments M30 and M65 and demonstrated high sensitivity for significant and advanced fibrosis detection [20]. The low specificity (<30%) indicates restricted ability for ruling in F2+ fibrosis at the defined cutoffs. For advanced fibrosis, CK-18 fragments demonstrated overall low accuracy (AUROC 0.590–0.600).

#### 3.3.2. Procollagen Type-III N-Terminal Peptide (PRO-C3)

Nielsen et al. reported that plasma PRO-C3 has satisfying diagnostic accuracy (AUROC = 0.700) for the detection of significant fibrosis among NASH patients with high specificity (86%) and rule-in potential [33]. For advanced fibrosis, AUROC of 0.730 was reported [33]. The cross-sectional study by Bril et al. identified PRO-C3 as one of the most reliable biomarkers for advanced fibrosis detection with a specificity of 96% and 0.900 AUROC at 20 ng/mL cutoff point [28].

#### 3.3.3. Monocyte Chemoattractant Protein 1 (MCP-1)

MCP-1 demonstrated modest diagnostic accuracy for the detection of significant and advanced fibrosis among NASH patients (AUROC of 0.520 and 0.510, respectively) [20]. In contrast, the high specificity of the MCP-1 biomarker reported in the same study indicated it could be reliable to rule in patients with significant advanced fibrosis (87% and 93%, respectively).

#### 3.3.4. NAFLD Fibrosis Protein Panel (NFPP) and a Disintegrin and Metalloproteinase with Thrombospondin Motifs like 2 (ADAMTSL2)

A retrospective study reported on two novel biomarkers for the detection of significant fibrosis among NAFLD patients, ADAMTSL2, and a combination of 8 sensitive proteins—NFPP [30]. Both biomarkers showed high diagnostic accuracy with AUROC of 0.830 for the detection of significant fibrosis. Additionally, the combination of NFPP with general clinical features (age, BMI, sex, and diabetes status), or with FIB-4 index or NFS improved the diagnostic accuracy of NFPP (AUROC 0.870) [30].

#### 3.3.5. Kulkarni Model

A large 10-year retrospective study of pediatric patients who underwent liver biopsy identified the strongest predictors of significant liver fibrosis. The model included body mass index, vitamin D, platelet count, and GGT and resulted in a very good predicting ability with sensitivity and specificity of more than 80% and AUROC of 0.944 [31].

#### 3.3.6. ADAPT Score

ADAPT score, based on the PRO-C3 levels, T2DM, platelet count, and age demonstrated satisfying accuracy in the detection of significant and advanced fibrosis among patients with definite NASH and liver fibrosis (AUROC 0.760 for F2+, 0.800 for F3+) [33].

#### 3.3.7. MEFIB Index

MEFIB index was determined using MRE with a cutoff value ≥ 3.3 kPa and FIB-4 index with a cutoff value ≥ 1.6 and provided a very high accuracy level for the detection of significant fibrosis [37]. Almost perfect specificity suggested this tool would be reliable to rule in NAFLD patients with significant fibrosis.

#### 3.3.8. FAST Score

FAST score combines liver stiffness measurement (LSM) and controlled attenuation parameter measured by VCTE (e.g., FibroScan) and serum levels of AST [38]. Overall, good accuracy of the FAST score was demonstrated in detecting definite NASH (NAFLD activity score ≥ 4 and significant fibrosis) among patients with NAFLD. A FAST score with lower cutoff values (0.35 and 0.38) demonstrated good ability to rule out F2+ [38].

#### 3.3.9. Cohort-Specific Model and Combination of 6 Biomarkers

The model included serum CK-18, fasting insulin, platelet count, sex, and HbA1c demonstrated good performance with an AUROC of 0.860 for advanced fibrosis detection among patients with T2DM [28]. A combination of 6 noninvasive tools (PRO-C3, APRI, AST, FIB-4 index, FibroTest, and NFS) showed very reliable performance with an AUROC of 0.910 in detecting advanced fibrosis among NAFLD patients with T2DM [28].

#### 3.3.10. Prognostic Factor Model

The model combining alkaline phosphatase, HbA1c, platelet count, and international normalized ratio performed well in detecting advanced fibrosis among NAFLD patients. However, high sensitivity and lower specificity indicated this noninvasive panel would be only reliable in ruling out patients with advanced fibrosis [29].

#### 3.3.11. Top 10 Metabolite Panel

The combination of 10-serum metabolites including lipids, amino acids, and carbohydrates (5α-androstane-3β monosulfate, pregnanediol-3-glucuronide, androsterone sulfate, epiandrosterone sulfate, palmitoleate, dehydroisoandrosterone sulfate, 5α-androstane-3β disulfate, glycocholate, taurine, and fucose) detected advanced fibrosis with high accuracy (AUROC 0.940) among NAFLD patients [29].

Accuracy of novel diagnostic tools is presented in Table 8 for significant fibrosis and Table 9 for advanced fibrosis.

## 4. Discussion

This SLR provides a comprehensive current overview of diagnostic tools for detection and monitoring of NASH-related liver fibrosis staging based on the summarized evidence from the US studies. The diagnostic accuracy was validated against liver biopsy as a standard diagnostic tool in all studies, except for the Caussy et al. study where both liver biopsy and MRE were used [29], and that of Younossi et al. where VCTE and biopsy were reference tools [36].

There is a remarkable shift in the diagnostic pathways from biopsy as the reference standard to novel, less invasive techniques, imaging methods, and blood biomarkers. Still, the collected evidence implies there is no perfect noninvasive tool capable of capturing and tracking all the aspects of the complex pathological process resulting in fatty liver, liver fibrosis, and cirrhosis. Liver biopsy often remains necessary in particular for clinical trials.

VCTE (FibroScan^®^) is a noninvasive ultrasound-based imaging method that measures the speed of passage of acoustic shear waves through the liver tissue to estimate liver stiffness. Our SLR provides collected evidence on good overall accuracy of VCTE. A prospective study conducted on liver transplant recipients by Siddiqui et al. reported high accuracy of LSM in the detection of significant fibrosis (AUROC of 0.870) and advanced fibrosis (AUROC of 0.940). Still, the PPV values lower than 60% indicate that the tool should be used carefully when ruling in the conditions [23]. Similar conclusions about the lower rule-in potential of VCTE were shown in a retrospective study by Trowell et al. [25]. Harrison et al. demonstrated a lower performance of VCTE in the detection of significant and advanced fibrosis among NASH patients (0.630 and 0.650, respectively) [20].

Another ultrasound-based imaging technique that demonstrated good accuracy with AUROC ranging from 0.730 to 0.850 was SWE for detecting significant and advanced fibrosis [22,26]. In the retrospective study conducted by Ozturk et al., SWE demonstrated good accuracy in a small sample of NAFLD patients with very advanced liver fibrosis [22]. Slightly better diagnostic accuracy of SWE in the detection of significant and advanced fibrosis was reported by Zhang et al. in their cross-sectional study conducted in the sample of 100 NAFLD patients (AUROC of 0.810 and 0.850, respectively) [26].

A promising imaging diagnostic method is MRE, which computes transversal images of liver tissue to capture the propagation of shear waves through the tissue. Results from our SLR are in correlation with the previously published data. A retrospective study conducted by Tang et al. reported very high diagnostic capabilities for the MRE technique in the detection of advanced fibrosis among NAFLD patients (AUROC of 0.939–0.947 at 3.6–3.65 kPa cutoffs). However, this study was conducted on a small sample of patients (19 patients) and the results should be further validated [39]. Zhang et al. compared diagnostic accuracy of MRE with SWE. Although MRE performed better in differentiating lower stages of fibrosis (F1+ and F2+), there was no difference between the tools in detecting F3+ fibrosis [40]. MRE was the only diagnostic tool captured in the SLR that tracked fibrosis improvement or progression in patients with F2 or F3 fibrosis stage. Even though the study had a small sample size and reported low sensitivity and specificity after 24 weeks from baseline fibrosis measurement, the authors concluded that MRE-LSM could potentially replace biopsy in evaluating longitudinal fibrosis changes [21].

The MRI-derived LSN score showed the lowest diagnostic accuracy for detecting significant and advanced fibrosis among all imaging techniques captured in the SLR when comparing the reported AUROCs. Despite a very high correlation between LSN score and level of fibrosis in overweight and obese patients with biopsy-proven NAFLD, AUROC values of 0.800 and 0.860 were the lowest compared to other imaging techniques [19].

Serum levels of liver enzymes AST and ALT are widely used blood biomarkers for the diagnosis of multiple conditions. Due to their low cost and wide availability, they are used to represent a starting point in fatty liver disease assessment [28]. However, the accuracy of AST and ALT levels in predicting significant and advanced fibrosis may be affected by other hepatic co-morbidities, patients’ characteristics, and associated conditions [41]. Accordingly, Harrison et al. demonstrated modest diagnostic accuracy among the population of patients diagnosed with NASH [20]. In contrast, Bril et al. suggest liver enzymes may remain the main diagnostic biomarker of advanced fibrosis in patients with T2DM due to their availability and high accuracy in excluding advanced cirrhosis, particularly in comparison with more costly and more complicated diagnostic options that turn out to perform equally well as AST/ALT levels in this population [28]. The authors recommended a sequential approach incorporating AST followed by another noninvasive tool for detecting advanced liver fibrosis, suggesting this approach would help avoid unnecessary liver biopsies [28].

Simple non-proprietary clinical scores (NFS, FIB-4, APRI) are cost-effective and sensitive enough to rule out the disease at lower thresholds. Still, they are not accurate enough to confirm the diagnosis of advanced fibrosis. BARD score and GGT were inferior in the detection of advanced fibrosis and showed modest accuracy with an AUROC in a range of 0.620–0.760 [20,27]. The Balakrishnan et al. study concluded that all investigated scores (NFS, APRI, BARD, FIB-4) have a moderate discriminatory ability for advanced fibrosis with AUROCs 0.700–0.790 [27] in predominantly Hispanic NAFLD patients. Similarly, the findings from Marella et al.’s study implied that noninvasive scores may be unreliable in the African American population and should be tested in larger multicenter studies [32]. Another study performed among T2DM patients with obesity also reported the modest accuracy of the scores in this population, with only FIB-4 showing a trend toward better accuracy [34]. Additionally, in a large observational study with more than two thousand patients, Udelsman et al. demonstrated low specificity of all noninvasive scoring systems in patients undergoing bariatric surgery [35]. Thus, although generally good accuracies imply the scoring systems can be reliable tools for the detection of significant and advanced fibrosis in everyday clinical practice, the modest performance in high-risk patients imposes the need for a more reliable screening assessment.

More expensive techniques that evaluate direct fibrosis markers (i.e., fibrosis markers in extracellular matrix components), such as the ELF test, CK-18 fragments, and PRO-C3, demonstrated higher sensitivity in detecting significant and advanced fibrosis. Based on our studies, these tests often cannot accurately differentiate progression or regression in diagnosed patients, and predominantly demonstrated modest diagnostic accuracy (AUROC < 0.8). Harrison et al. reported that despite suboptimal performance of noninvasive biomarkers in general, ELF demonstrated somewhat better diagnostic ability in fibrosis detection [20]. A large retrospective cohort analysis by Younossi et al. emphasized that ELF may be a very valuable tool for advanced fibrosis detection with high NPV and PPV but using multiple cohort-specific cutoff values that need to be validated before the use in clinical practice [36].

The FibroTest demonstrated modest diagnostic accuracy in the detection of advanced fibrosis among T2DM patients [28]. The ADAPT score showed more promising results in the detection of significant and advanced fibrosis within the sample of patients with NASH and liver fibrosis [33]. The regression model developed by Kulkarni et al. demonstrated very high accuracy in the detection of significant fibrosis among NAFLD patients [31]. Still, the retrospective nature of the data and the lack of prospective validation prevent us from concluding about the potential utility of the combined biomarkers.

In recent years, several specific metabolomic profiles have been associated with different stages of disease in NAFLD patients, making them a good target for future research in NAFLD diagnostics. Further studies revealed that changes in levels of these metabolites additionally could reflect specific pathways of liver injury related to NASH or advanced fibrosis, making them compelling diagnostic biomarkers. Therefore, it has been suggested that a combination of blood metabolites could be a highly accurate diagnostic test for the detection of advanced fibrosis [29]. Furthermore, many researchers evaluated the diagnostic potential of different combinations of biomarkers and diagnostic scores for fibrosis detection to achieve greater accuracy and better prediction power.

Caussy et al. demonstrated that a combination of 10 serum metabolites including lipids, amino acids, and carbohydrates had a very good discriminatory ability for the detection of advanced fibrosis among patients with biopsy-proven NAFLD [29]. The specific panel of blood biomarkers showed greater diagnostic accuracy with higher AUROC values than the FIB-4 Index and NFS to detect advanced fibrosis, which was confirmed afterwards in two independent validation cohorts. Moreover, the panel demonstrated the ability to evaluate longitudinal changes in serum metabolites in assessing the disease progression, which is a valuable characteristic rarely seen among biomarkers [29]. Harrison et al. assessed the diagnostic accuracy of MCP-1 and liver fibrosis-specific protein [20], and Corey et al. evaluated the diagnostic accuracy of NFPP and ADAMTSL2 in the detection of significant and advanced fibrosis, aiming to detect the “protein-based signature of fibrosis” [30]. MCP-1 demonstrated a low level of diagnostic accuracy for detecting significant and advanced fibrosis among NASH patients [20], while NFPP and ADAMTSL2 showed promising results in the detection of advanced fibrosis among NAFLD patients (AUROC of 0.830) [30]. Decraecker et al. demonstrated in metabolic (dysfunction)-associated fatty liver disease (MAFLD) patients with liver stiffness measurements, FIB-4, and LIVERFASt, that noninvasive methods were correlated with overall and liver-related mortalities (*p* < 0.001), and with all-cause and liver-related outcomes (*p* < 0.001) [42].

Studies captured in this SLR provide an insight into the new perspectives on diagnostic tools and panels which combined imaging techniques and blood-based biomarkers for detecting significant fibrosis. The FAST score is a novel technique that combines liver stiffness measurement and controlled attenuation parameters measured by VCTE and serum levels of AST. That score was already established in a European cohort and Woreta et al. validated the results in the US population [38]. The FAST score demonstrated high diagnostic accuracy in the detection of significant fibrosis among NAFLD patients. Still, the modest PPV implies the score should be interpreted carefully when ruling in patients with significant fibrosis [38]. The other novel technique established by Jung et al. demonstrated even better diagnostic accuracy of the MEFIB index, and MRE liver examination in combination with the FIB-4 index, for detecting significant fibrosis among NAFLD patients [37].

Our findings are in line with the results of other studies. Chalasani et al. reported that 27% of patients evaluated with VCTE yielded unreliable results [43] while a European retrospective study demonstrated the high accuracy of VCTE with an AUROC of 0.800 at 9.9 kPa and 11.4 kPa cutoff values for the detection of advanced fibrosis among patients with biopsy-proven NASH [44]. A previously published meta-analysis comparing the accuracy of VCTE and MRE in fibrosis detection concluded that MRE provides significantly greater accuracy, although both methods performed very well in NAFLD patients [45]. However, despite the good reliability of MRE in detecting and differentiating liver fibrosis, the decision to use one method over another depends on multiple factors, including the availability of the tool and cost-effectiveness. MRE also requires special equipment, software, additional hardware beyond routine scanners, as well as experienced experts for results validation and interpretation [46]. Thus, it is unlikely that MRE will replace US-based imaging methods for the detection and longitudinal tracking of liver fibrosis in routine clinical practice in the near future. The overarching pitfall of all imaging methods is unreliable specificity and dependence on the screener’s experience and subjectivity in determining the fibrosis stage. Furthermore, imaging techniques may be non-standardized, costly, not widely available, or inaccessible [45].

Similarly, although multiple noninvasive biomarkers are available on the market for detecting significant or advanced liver fibrosis, there is still an unmet need for a test that would provide more accurate staging and differentiation of fibrosis as currently used noninvasive tests remain inconclusive in approximately 30% of patients [47]. The Noninvasive Biomarkers of Metabolic Liver Disease (NIMBLE) consortium demonstrated that only NIS4, ELF test, and FibroMeter-VCTE met the predefined criteria (AUROC > 0.800) for accurate diagnosis of significant fibrosis (F2+ stages), while the ELF test and FibroMeter-VCTE met criteria for successful determination of advanced fibrosis (F3+ stages). Other investigated tests (FIB-4 index, serum ALT levels, OWLiver, and serum PRO-C3 levels) did not satisfy the criteria in terms of diagnostic accuracy in the detection of F2+ and F3+ fibrosis stages [48]. No diagnostic test addressed the unmet need in terms of diagnostic tool sensitivity and specificity of >80% in the detection of any stage of fibrosis, which was the minimum acceptable level specified by payers in the US. Of all evaluated diagnostic panels, only NIS4 for the detection of significant fibrosis and FibroMeter-VCTE for the detection of advanced fibrosis met the criteria specified by the US healthcare providers (sensitivity and specificity > 75%) [48]. The European Association for the Study of the Liver (EASL) clinical practice guidelines for NASH listed several reliable biomarkers for fibrosis detection with AUROC values higher than 0.8, including NFS, FIB-4 index, ELF, and FibroTest [49,50]. Apart from fibrosis detection, the tests also predicted liver-related and overall mortality with good precision. Still, the guidelines emphasize that the tests can correctly distinguish advanced fibrosis from lower stages, but not significant fibrosis. Additionally, the high NPV show that the tests perform particularly well in excluding advanced fibrosis so that they can be used as a first-line strategy in risk stratification, while they are not as good in ruling in fibrosis [49]. The study points out that predictive values are highly dependent on fibrosis prevalence in the study population, which is generally higher than in the community. Thus, the generalizability of the study findings would need to be tested in larger populations of patients [49]. A recently published review reports on the use of serum fibrosis biomarkers based on routine biochemistry and VCTE as validated and well-incorporated screening strategies for identifying high-risk patients. Still, MRI techniques are seen as the most promising noninvasive diagnostic strategy as they offer accurate fibrosis staging with the ability to assess therapeutic response [51]. Finally, a systematic review of available guidelines for NAFLD assessment concludes that fibrosis scores may help detect high-risk patients who may be referred to liver biopsy; still, all guidelines stress the necessity of developing a noninvasive test that will replace liver biopsy as a research priority [52].

Liver biopsy remains the reference standard in fibrosis detection and classification [53,54]. However, a growing body of evidence highlights the limitations of liver biopsy, particularly in fibrosis detection [55]. Aside from the risk of complications and invasive nature, sampling variability remains a big concern as histological lesions are unevenly distributed throughout the liver tissue [56]. Further problems with pathological diagnosis arise with inter- and intra-observer variability. Therefore, evaluating test accuracy with an imperfect reference standard such as a liver biopsy poses the risk of underestimating NASH and fibrosis severity [55].

### Study Strengths and Limitations

The major strength of this review is that this is the first study to systematically summarize and compare noninvasive diagnostic tools evaluated in the US healthcare context. As the majority of noninvasive diagnostic tools were compared to the biopsy gold standard, this review collects clinically relevant evidence, establishing basis for decision making in the field. This SLR has several limitations. As in all literature reviews, it has to be acknowledged that the reliability of the findings depends on the methodology and validity of the primary studies. Only diagnostic tools with either sensitivity or specificity values were presented in the Results section. The SLR considered only two literature databases, PubMed and Web of Science. Another limitation is the heterogeneity of the included studies that prevented us from synthesizing quantitative evidence and providing narrow point estimates of the effective measures. This review aimed to systematically summarize the recent trends in clinical practice and diagnostic research in NAFLD, identifying articles published from 2016 onwards. It has to be denoted that the review may have omitted promising research tools published prior to 2016. Thus, the study results were discussed and interpreted with caution, paying particular attention to the existing body of evidence and ensuring the conclusions are in line with findings from previous systematic literature reviews [10,47,57,58]. Some of the included studies were conducted in small population samples; therefore, the demonstrated results may lack generalizability. Additionally, some diagnostic tools were assessed in only one published article, so the findings have to be interpreted with caution until confirmed in larger observational studies. Furthermore, the lack of direct head-to-head comparisons between the diagnostic strategies limits the possibility of unbiased comparison of diagnostic accuracy between the tools. Our study concentrated on the diagnostic ability of noninvasive tools to identify and differentiate significant and advanced fibrosis (F2+ and F3+), stipulating that the diagnosis made at this stage may impact clinical decision-making and change the course of the disease. Still, the efficiency of the tools in detecting earlier stages of fibrosis (F1+) has not been reviewed, while it may be increasingly important in the disease course, as treatment measures at earlier stages may positively impact the disease outcomes [59]. Additionally, some of the multicenter studies cited here did not have a centralized pathological reading, which then introduces substantial bias in relation to inter-pathologist variability. Finally, quality assessment and critical appraisal of the studies were not performed. The SLR presents only formally published data, which may lead to publication bias, as journals are strongly biased towards publishing only the studies that report significant differences in the results. Still, as the SLR was not primarily focused on treatment effectiveness, the probability of bias in our review is low.

## 5. Conclusions

Liver fibrosis detection, staging, and monitoring represent crucial points in the clinical assessment and risk evaluation of patients with nonalcoholic liver disease, as it is the major predictor of patients’ morbidity and mortality. Imaging techniques represent an important part of the management of patients with suspected liver fibrosis, becoming increasingly incorporated into routine clinical practice. Imaging techniques overcome limitations of liver biopsy, such as discomfort, invasiveness, and repeated tissue sampling, providing good overall accuracy in fibrosis detection. However, they are still not able to provide a sufficiently sensitive and reliable estimate of quantitative longitudinal and dynamic changes in fibrosis development and regression. Moreover, advanced imaging techniques like VCTE and MRE require costly equipment and trained personnel, so they are less available in clinical practices across the country.

Observing the wide spectrum of available biomarkers, including clinical scores and panels that combine several blood tests, it may be correctly concluded that noninvasive biomarkers play a significant and ever-increasing role in detecting liver fibrosis in patients with NAFLD, including high-risk subgroups of patients. On the other hand, observing the high sensitivity, specificity, and AUROC values presented in the studies, it could be falsely concluded that there are tools with almost perfect diagnostic abilities that may detect liver fibrosis with a precision equivalent to substantially more expensive imaging methods or even biopsy. Still, in reality, the perfect noninvasive biomarker has not been established so far. In general, noninvasive biomarkers demonstrated very good accuracy in excluding significant or advanced liver fibrosis. The highest accuracy was observed among scores that combine several biomarkers, metabolites, and clinical parameters. Still, the studies generally conclude that all these tests have limited power in detecting and quantifying fibrosis levels, which is necessary for patient management and monitoring of disease progression. Several innovative technologies that demonstrated promising initial results in small patient cohorts have to be externally validated in wider independent studies. There is still an unmet need for a noninvasive biomarker that can detect, measure, and differentiate fibrosis stages with great sensitivity and specificity. Additionally, the optimal diagnostic tool has to be easily applicable and affordable for patients, providers, and healthcare centers at all levels, due to the rising prevalence of the disease among all age categories, ethnicities, and risk groups. Nonetheless, because of the variability of the biopsy itself as well as of the pathological reading, the ultimate validation of noninvasive markers will involve their ability to predict clinical events rather than a particular histological lesion.

## Figures and Tables

**Figure 1 diagnostics-12-02608-f001:**
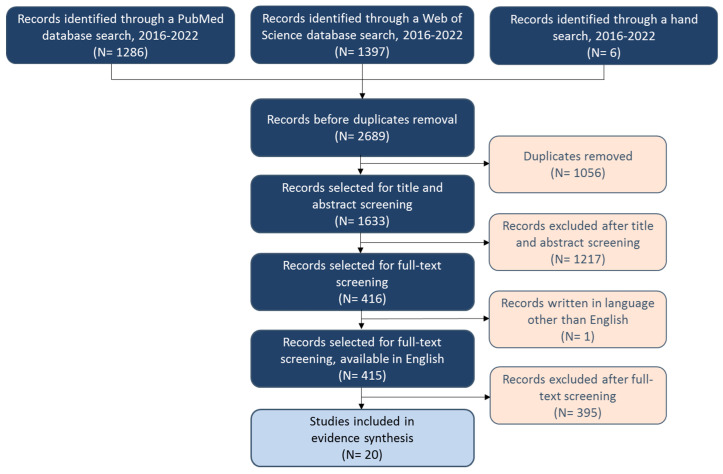
PRISMA flow diagram.

**Figure 2 diagnostics-12-02608-f002:**
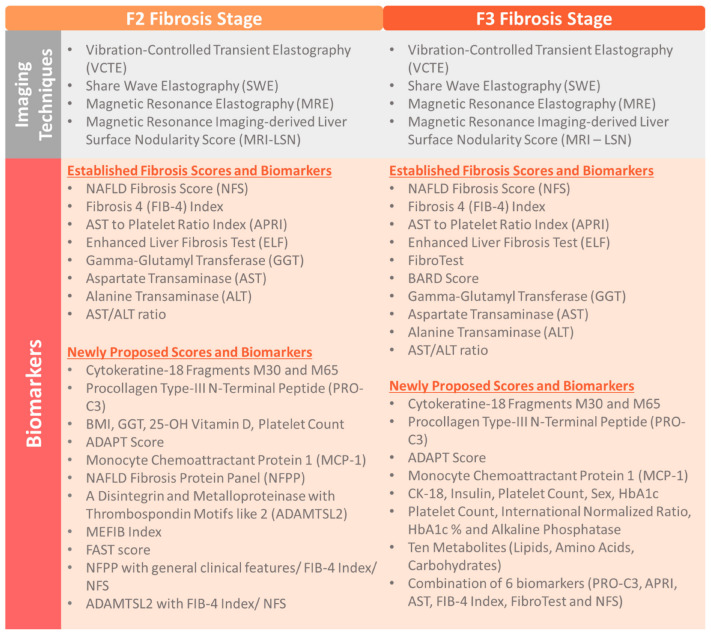
Summary of diagnostic modalities obtained in the SLR.

**Table 1 diagnostics-12-02608-t001:** Selection criteria.

Inclusion Criteria	Exclusion Criteria
1. Published between January 2016 and May 2022 2. Written in English 3. Human studies 4. Original research	1. Studies with irrelevant outcomes 2. No full-text studies 3. In vitro studies 4. Molecular and genetic studies 5. Editorials, comments, replies, and letters to the author

**Table 2 diagnostics-12-02608-t002:** PICO criteria for diagnostic tools accuracy.

PICO	Inclusion Criteria	Exclusion Criteria
Population	1. Patients diagnosed with NAFLD 2. Patients diagnosed with NASH 3. Patients diagnosed with significant or advanced liver fibrosis 4. Patients diagnosed with significant or severe liver steatosis 5. Patients diagnosed with liver cirrhosis	1. Nonhuman population
Interventions	1. Any treatment or management 2. No treatment or management	NA
Comparators	1. Any treatment or management 2. No treatment or management	NA
Outcomes	1. Sensitivity 2. Specificity 3. Positive predictive value 4. Negative predictive value 5. Area under the receiver operating characteristic curve	NA
Restrictions	1. English language 2. Year limitation: 2016–2022	1. Genetic studies 2. Editorials 3. Letters and comments to the authors 4. Case reports 5. SLRs, meta-analyses, narrative reviews, guidelines

NAFLD—nonalcoholic fatty liver disease; NASH—nonalcoholic steatohepatitis; NA—not applicable; SLR—systematic literature review.

**Table 3 diagnostics-12-02608-t003:** Characteristics of included studies.

**Studies That Evaluated Accuracy of Imaging Techniques**						
**Author**	**Year**	**Study Type**	**Population**	**Baseline Fibrosis**	**Imaging Technique**	**Detection Capabilities**
Catania et al. [19]	2021	Prospective study	47 NAFLD patients	Fibrosis stage 2: 39% Fibrosis stage 3: 25% Fibrosis stage 4: 19%	MRI-derived LSN score	Significant fibrosis Advanced fibrosis
Harrison et al. [20]	2020	Prospective study	288 NASH patients with F1-F3 fibrosis	FIB-4 score, mean (SD): 1.4 (59.3)	VCTE	Significant fibrosis Advanced fibrosis
Jayakumar et al. [21]	2019	Prospective study	54 NASH patients with F2-F3 fibrosis	Fibrosis stage 2: 37%Fibrosis stage 3: 67%	MRE	Significant fibrosis Advanced fibrosis (progression and improvement)
Ozturk et al. [22]	2020	Retrospective study	116 NAFLD patients	Fibrosis stage 2: 9.4%Fibrosis stage 3: 13.7%Fibrosis stage 4: 2.5%	SWE	Significant fibrosis Advanced fibrosis
Siddiqui et al. [23]	2021	Prospective study	99 patients with liver transplantation history	Fibrosis stage 2: 4.0% Fibrosis stage 3: 10.1% Fibrosis stage 4: 7.1%	VCTE	Significant fibrosis Advanced fibrosis
Tang et al. [24]	2022	Retrospective study	91 NAFLD patients	Fibrosis stage 2: 7.7% Fibrosis stage 3: 11% Fibrosis stage 4: 5.5%	MRE	Advanced fibrosis
Trowell et al. [25]	2021	Retrospective study	217 NAFLD and non-NAFLD patients	Fibrosis stage 2: 24% Fibrosis stage 3: 13% Fibrosis stage 4: 18%	VCTE	Advanced fibrosis
Zhang et al. [26]	2022	Cross-sectional study	100 NAFLD patients	Fibrosis stage 2: 5% Fibrosis stage 3: 10% Fibrosis stage 4: 6%	SWE, MRE	Significant fibrosis Advanced fibrosis
**Studies that evaluated accuracy of established fibrosis scores and biomarkers**						
**Author**	**Year**	**Study Type**	**Population**	**Baseline Fibrosis**	**Biomarker**	**Detection Capabilities**
Balakrishnan et al. [27]	2021	Retrospective cross-sectional study	99 NAFLD patients	Fibrosis stage 0–2: 62.6% Fibrosis stage 3–4: 37.4% FIB-4 score, mean (SD): In fibrosis stage 0–2: 0.99 (0.55) In fibrosis stage 3–4: 2.23 (1.52)	NFS, FIB-4 index, APRI, BARD score	Significant fibrosis
Bril et al. [28]	2020	Cross-sectional study	213 T2DM patients	Fibrosis stage, mean (SD): In no NASH: 0.6 (0.9) In definite NASH: 1.8 (1.0)	NFS, FIB-4 index, APRI, Plasma AST levels FibroTest	Advanced fibrosis
Caussy et al. [29]	2019	Cross-sectional study	156 NAFLD patients	FIB-4 score, mean (SD): 1.35 (1.24)	NFS, FIB-4 index	Advanced fibrosis
Corey et al. [30]	2022	Retrospective chart review	84 NAFLD patients	Fibrosis stage 2: 25% Fibrosis stage 3: 14% Fibrosis stage 4: 10%	NFS, FIB-4 index	Significant fibrosis
Harrison et al. [20]	2020	Prospective study	288 NASH patients	FIB-4 score, mean (SD): 1.4 (59.3)	NFS, FIB-4 index, Plasma AST levels, Plasma ALT levels, GGT levels ELF test	Significant fibrosisAdvanced fibrosis
Kulkarni et al. [31]	2021	Retrospective study	55 NAFLD patients	Fibrosis stage 2: 20% Fibrosis stage 3: 7.3% Fibrosis stage 4: 3.6%	GGT levels	Significant fibrosis
Marella et al. [32]	2020	Retrospective chart review	907 NAFLD patients	Fibrosis stage 2: 17.9% Advanced fibrosis: 12.8% Fibrosis score, mean (SD): 1.16 (1.13) FIB-4 score, mean (SD): 1.28 (1.75)	NFS, FIB-4 index, APRI	Advanced fibrosis
Nielsen et al. [33]	2021	Retrospective database study	517 patients with NASH and fibrosis	Fibrosis stage 2: 21% Fibrosis stage 3: 24% Fibrosis stage 4: 5%	FIB-4 index, APRI, AST/ALT ratio	Significant fibrosisAdvanced fibrosis
Singh et al. [34]	2020	Retrospective chart review	1157 adult diabetics with NAFLD	Fibrosis stage 0–2: 68% Fibrosis stage 3–4: 32%	NFS, FIB-4 index, APRI, AST/ALT ratio	Advanced fibrosis
Udelsman et al. [35]	2021	Retrospective chart review	2465 patients	Fibrosis stage 3+: 3.4%	NFS, FIB-4 index, APRI	Advanced fibrosis
Younossi et al. [36]	2021	Retrospective cross-sectional study	829 NAFLD patients	FIB-4 score, mean (SD): 1.34 (0.97)	ELF test	Advanced fibrosis
**Studies that evaluated accuracy of novel biomarkers**						
**Author**	**Year**	**Study Type**	**Population**	**Baseline Fibrosis**	**Diagnostic Technique**	**Detection Capabilities**
Corey et al. [30]	2022	Retrospective chart review	84 NAFLD patients	Fibrosis stage 2: 25% Fibrosis stage 3: 14% Fibrosis stage 4: 10%	NFPP, ADAMTSL2, and these in combination with general clinical features, FIB-4 index, or NFS	Significant fibrosis
Bril et al. [28]	2020	Cross-sectional study	213 T2DM patients	Fibrosis stage, mean (SD): In no NASH: 0.6 (0.9) In definite NASH: 1.8 (1.0)	PRO-C3, Cohort-specific model, Combination of 6 biomarkers	Advanced fibrosis
Caussy et al. [29]	2019	Cross-sectional study	156 NAFLD patients	FIB-4 score, mean (SD): 1.35 (1.24)	Prognostic factor model, Top 10 metabolite panel	Advanced fibrosis
Harrison et al. [20]	2020	Prospective study	288 NASH patients	FIB-4 score, mean (SD): 1.4 (59.3)	CK-18 fragment M30, CK-18 fragment M65, MCP-1	Significant fibrosisAdvanced fibrosis
Kulkarni et al. [31]	2021	Retrospective study	55 NAFLD patients	Fibrosis stage 2: 20% Fibrosis stage 3: 7.3% Fibrosis stage 4: 3.6%	Scoring system	Significant fibrosis
Nielsen et al. [33]	2021	Retrospective database study	517 patients with NASH and fibrosis	Fibrosis stage 2: 21% Fibrosis stage 3: 24% Fibrosis stage 4: 5%	PRO-C3, ADAPT score	Significant fibrosisAdvanced fibrosis
Jung et al. [37]	2021	Prospective study	238 NAFLD patients	Fibrosis stage 2: 11.3% Fibrosis stage 3: 9.7% Fibrosis stage 4: 7.6% FIB-4 score, mean (SD): 1.5 (1.4)	MEFIB index	Significant fibrosis
Woreta et al. [38]	2022	Retrospective study	585 NAFLD patients	Fibrosis stage 2: 20.6% Fibrosis stage 3: 20.7% Fibrosis stage 4: 10.4%	FAST score	Significant fibrosis

**Table 4 diagnostics-12-02608-t004:** Diagnostic accuracy of imaging techniques for significant fibrosis.

**Vibration-Controlled Transient Elastography**						
Source	Cutoff	Sensitivity	Specificity	PPV	NPV	AUROC
Harrison et al. [20]	7.3 kPa	89.0%	33.0%	77.0%	56.0%	0.630
Siddiqui et al. [23]	7.4 kPa	90.0%	60.0%	38.0%	96.0%	0.870
	10.5 kPa	81.0%	83.0%	57.0%	94.0%	
	13.5 kPa	67.0%	90.0%	67.0%	91.0%	
**Shear Wave Elastography**						
Source	Cutoff	Sensitivity	Specificity	PPV	NPV	AUROC
Ozturk et al. [22]	8.4 kPa	77.0%	66.0%	-	-	0.730
Zhang et al. [26]	1.49 m/s	90.5%	43.0%	29.7%	94.4%	0.810
	1.79 m/s	47.6%	91.1%	58.8%	86.7%	
**Magnetic Resonance Elastography**						
Source	Cutoff	Sensitivity	Specificity	PPV	NPV	AUROC
Zhang et al. [26]	2.77 kPa	90.5%	84.8%	61.3%	97.1%	0.940
	3.06 kPa	81.0%	91.1%	70.8%	94.7%	
**Magnetic Resonance Imaging-Derived Liver Surface Nodularity Score**						
Source	Cutoff	Sensitivity	Specificity	PPV	NPV	AUROC
Catania et al. [19]	2.23	72.0%	62.0%	-	-	0.800

Abbreviations: PPV—positive predictive value; NPV—negative predictive value; AUROC—area under the receiver operating characteristic curve.

**Table 5 diagnostics-12-02608-t005:** Diagnostic accuracy of imaging techniques for advanced fibrosis.

**Vibration-Controlled Transient Elastography**						
Source	Cutoff	Sensitivity	Specificity	PPV	NPV	AUROC
Harrison et al. [20]	11.5 kPa	56.0%	71.0%	65.0%	63.0%	0.650
Siddiqui et al. [23]	10.5 kPa	94.0%	83.0%	53.0%	99.0%	0.940
	10.5 kPa	90.0%	83.0%	53.0%	99.0%	
	13.3 kPa	82.0%	90.0%	64.0%	96.0%	
Trowell et al. [25]	11.9 kPa ^1^	75.0%	81.5%	65.4%	87.5%	0.850
	11.9 kPa ^2^	73.7%	74.5%	53.8%	87.5%	0.780
**Shear Wave Elastography**						
Source	Cutoff	Sensitivity	Specificity	PPV	NPV	AUROC
Ozturk et al. [22]	8.4 kPa	84.0%	70.0%	-	-	0.820
Zhang et al. [26]	1.46 m/s	93.8%	39.3%	39.3%	97.1%	0.850
	1.78 m/s	62.5%	90.5%	55.6%	92.7%	
**Magnetic Resonance Elastography**						
Source	Cutoff	Sensitivity	Specificity	PPV	NPV	AUROC
Tang et al. [24]	3.6 kPa ^3^	93.0%	95.0%	78.0%	99.0%	0.939
	3.65 kPa ^4^	93.0%	95.0%	78.0%	99.0%	0.947
	3.65 kPa ^5^	93.0%	93.0%	74.0%	99.0%	0.940
Zhang et al. [26]	2.77 kPa	93.8%	81.0%	81.0%	98.6%	0.950
	3.17 kPa	81.3%	90.5%	61.9%	96.2%	
**Magnetic Resonance Imaging-Derived Liver Surface Nodularity Score**						
Source	Cutoff	Sensitivity	Specificity	PPV	NPV	AUROC
Catania et al. [19]	2.44	81.0%	88.0%	-	-	0.860

Abbreviations: PPV—positive predictive value; NPV—negative predictive value; AUROC—area under the receiver operating characteristic curve. ^1^ Reported results are obtained from the training cohort. ^2^ Reported results are obtained from the validation cohort. ^3^ Diagnostic accuracy at Center 1. ^4^ Diagnostic accuracy at Center 2. ^5^ Diagnostic accuracy of automated liver stiffness analysis.

**Table 6 diagnostics-12-02608-t006:** Diagnostic accuracy of established scores and biomarkers in detecting significant fibrosis.

**NAFLD Fibrosis Score**						
Source	Cutoff	Sensitivity	Specificity	PPV	NPV	AUROC
Corey et al. [30]	-	36%	85%	67%	62%	0.640
Harrison et al. [20]	0.9	66%	52%	77%	38%	0.600
**Fibrosis-4 index**						
Source	Cutoff	Sensitivity	Specificity	PPV	NPV	AUROC
Corey et al. [30]	-	48%	88%	76%	68%	0.700
Harrison et al. [20]	1.3	64%	70%	84%	44%	0.690
Nielsen et al. [33]	>1.12	71%	62%	65%	69%	0.710
**AST to Platelet Ratio Index**						
Source	Cutoff	Sensitivity	Specificity	PPV	NPV	AUROC
Nielsen et al. [33]	>0.42	57%	67%	63%	61%	0.660
**Enhanced Liver Fibrosis test**						
Source	Cutoff	Sensitivity	Specificity	PPV	NPV	AUROC
Harrison et al. [20]	−0.2	62%	68%	83%	42%	0.690
**GGT levels**						
Source	Cutoff	Sensitivity	Specificity	PPV	NPV	AUROC
Harrison et al. [20]	70.0 U/L	40%	72%	79%	32%	0.560
Kulkarni et al. [31]	65 U/L	66%	76%	-	-	-
**AST/ALT ratio**						
Source	Cutoff	Sensitivity	Specificity	PPV	NPV	AUROC
Nielsen et al. [33]	>0.56	90%	25%	54%	71%	0.580
**AST levels**						
Source	Cutoff	Sensitivity	Specificity	PPV	NPV	AUROC
Harrison et al. [20]	42.0 U/L	57%	68%	82%	38%	0.630
**ALT levels**						
Source	Cutoff	Sensitivity	Specificity	PPV	NPV	AUROC
Harrison et al. [20]	54.0 U/L	53%	60%	77%	33%	0.550

Abbreviations: NAFLD—nonalcoholic fatty liver disease; PPV—positive predictive value; NPV—negative predictive value; AUROC—area under the receiver operating characteristic curve; GGT—gamma-glutamyl transferase; AST—aspartate transaminase; ALT—alanine transaminase.

**Table 7 diagnostics-12-02608-t007:** Diagnostic accuracy of established scores and biomarkers for detecting advanced fibrosis.

**NAFLD Fibrosis Score**						
Source	Cutoff	Sensitivity	Specificity	PPV	NPV	AUROC
Balakrishnan et al. [27]	≥−1.455	81.1%	66.1%	58.8%	85.4%	0.790
	≥0.676	32.4%	95.2%	80.0%	70.2%	
Bril et al. [28]	<−1.455 and >0.676	91%	40%	26%	95%	0.640
	−0.053	68%	55%	21%	90%	
Caussy et al. [29]	-	90%	59%	28%	97%	0.840
Harrison et al. [20]	0.9	71%	48%	57%	63%	0.580
Marella et al. [32]	>0.675	57%	84%	35%	93%	0.810
Singh et al. [34]	>0.676	63.7%	70%	49.8%	80.5%	0.720
	≥(−1.455)	94.6%	16.9%	34.7%	87.1%	
Udelsman et al. [35]	<−1.455	85%	38%	5%	99%	0.720
	>0.675	40%	85%	9%	98%	
**Fibrosis-4 Index**						
Source	Cutoff	Sensitivity	Specificity	PPV	NPV	AUROC
Balakrishnan et al. [27]	≥1.3	56.8%	77.4%	60%	75%	0.770
	≥2.67	40.5%	100%	100%	73.8%	
Bril et al. [28]	<1.45 and >3.25	33%	99%	80%	94%	0.780
	1.666	68%	75%	31%	93%	
Caussy et al. [29]	-	90%	39%	21%	96%	0.780
Harrison et al. [20]	1.3	69%	64%	65%	68%	0.670
Marella et al. [32]	> 2.67	29%	98%	66%	90%	0.880
Nielsen et al. [33]	>1.12	87%	59%	46%	92%	0.790
Singh et al. [34]	>2.67	44.1%	93%	74.5%	78.3%	0.770
	≥1.45	72.6%	64.4%	48.5%	83.6%	
Udelsman et al. [35]	>1.30	58%	86%	13%	98%	0.790
	>2.67	21%	99%	55%	97%	
**AST to Platelet Ratio Index**						
Source	Cutoff	Sensitivity	Specificity	PPV	NPV	AUROC
Balakrishnan et al. [27]	≥1	48.7%	88.7%	72%	74.3%	0.700
Bril et al. [28]	<0.5 and >1.5	31%	99%	67%	94%	0.860
	0.423	84%	75%	36%	96%	
Marella et al. [32]	>1.5	14%	98%	47%	89%	0.830
Nielsen et al. [33]	>0.34	79%	51%	39%	86%	0.680
Singh et al. [34]	>1.5	16.5%	97.4%	74.7%	71.7%	0.740
	≥1	27.9%	94.7%	70.9%	74%	
Udelsman et al. [35]	>0.98	24%	99%	65%	97%	0.810
**BARD Score**						
Source	Cutoff	Sensitivity	Specificity	PPV	NPV	AUROC
Balakrishnan et al. [27]	≥2	75.7%	59.7%	52.8%	80.4%	0.760
**Enhanced Liver Fibrosis Test**						
Source	Cutoff	Sensitivity	Specificity	PPV	NPV	AUROC
Harrison et al. [20]	−0.1	67%	63%	63%	66%	0.680
Younossi et al. [36]	9.8 ^1^	57.5%	88.9%	62.5%	88.6%	0.810
	11.3 ^1^	19.5%	99.1%	88.0%	79.2%	
	9.8 ^2^	58.2%	84.1%	43.0%	90.7%	0.790
	11.3 ^2^	17.7%	99.5%	87.5%	85.4%	
**FibroTest**						
Source	Cutoff	Sensitivity	Specificity	PPV	NPV	AUROC
Bril et al. [28]	<0.3 and >0.7	17.0%	98.0%	40.0%	92.0%	0.700
	0.353	64.0%	74.0%	30.0%	92.0%	
**GGT Levels**						
Source	Cutoff	Sensitivity	Specificity	PPV	NPV	AUROC
Harrison et al. [20]	68.0 U/L	49%	72%	63%	59%	0.620
**AST/ALT Ratio**						
Source	Cutoff	Sensitivity	Specificity	PPV	NPV	AUROC
Nielsen et al. [33]	>0.78	63%	64%	42%	81%	0.680
Singh et al. [34]	>1.4	27.4%	84.2%	44.6%	71.5%	0.620
	≥1	60.7%	53.3%	37.6%	74.5%	
**AST Levels**						
Source	Cutoff	Sensitivity	Specificity	PPV	NPV	AUROC
Bril et al. [28]	40 U/L	77%	81%	41%	96%	0.850
	38 U/L	84%	79%	40%	97%	
Harrison et al. [20]	37 U/L	73%	52%	60%	67%	0.660
**ALT Levels**						
Source	Cutoff	Sensitivity	Specificity	PPV	NPV	AUROC
Harrison et al. [20]	68.0 U/L	41%	74%	61%	56%	0.580

Abbreviations: NAFLD—nonalcoholic fatty liver disease; PPV—positive predictive value; NPV—negative predictive value; AUROC—area under the receiver operating characteristic curve; AST—aspartate transaminase; ALT—alanine transaminase; BARD score—BMI. AST, ALT, and diabetes mellitus presence; GGT—gamma-glutamyl transferase; BMI—body-mass index; ^1^ biopsy was used as a reference tool; ^2^ vibration-controlled transient elastography was used as a reference tool.

**Table 8 diagnostics-12-02608-t008:** Diagnostic performance of novel biomarkers in the detection of significant fibrosis.

**Cytokeratine-18 Fragment M30**						
Source	Cutoff	Sensitivity	Specificity	PPV	NPV	AUROC
Harrison et al. [20]	260 U/L	90%	26%	76%	50%	0.560
**Cytokeratine-18 Fragment M65**						
Source	Cutoff	Sensitivity	Specificity	PPV	NPV	AUROC
Harrison et al. [20]	545 U/L	90%	29%	77%	54%	0.580
**Procollagen Type-III N-Terminal Peptide**						
Source	Cutoff	Sensitivity	Specificity	PPV	NPV	AUROC
Nielsen et al. [33]	19.65 ng/mL	45%	86%	76%	61%	0.700
**Monocyte Chemoattractant Protein 1**						
Source	Cutoff	Sensitivity	Specificity	PPV	NPV	AUROC
Harrison et al. [20]	497.2	21%	87%	80%	30%	0.520
**NAFLD Fibrosis Protein Panel**						
Source	Cutoff	Sensitivity	Specificity	PPV	NPV	AUROC
Corey et al. [30]	-	64%	86%	78%	76%	0.830
**NAFLD Fibrosis Protein Panel with General Clinical Features**						
Source	Cutoff	Sensitivity	Specificity	PPV	NPV	AUROC
Corey et al. [30]	-	70%	93%	88%	80%	0.870
**NAFLD Fibrosis Protein Panel and FIB-4 Index**						
Source	Cutoff	Sensitivity	Specificity	PPV	NPV	AUROC
Corey et al. [30]	-	73%	85%	80%	80%	0.870
**NAFLD Fibrosis Protein Panel and NFS**						
Source	Cutoff	Sensitivity	Specificity	PPV	NPV	AUROC
Corey et al. [30]	-	76%	85%	81%	81%	0.870
**A Disintegrin and Metalloproteinase with Thrombospondin Motifs like 2**						
Source	Cutoff	Sensitivity	Specificity	PPV	NPV	AUROC
Corey et al. [30]	-	58%	91%	83%	74%	0.830
**A Disintegrin and Metalloproteinase with Thrombospondin Motifs like 2 with FIB-4**						
Source	Cutoff	Sensitivity	Specificity	PPV	NPV	AUROC
Corey et al. [30]	-	67%	85%	79%	76%	0.830
**A Disintegrin and Metalloproteinase with Thrombospondin Motifs like 2 with NFS**						
Source	Cutoff	Sensitivity	Specificity	PPV	NPV	AUROC
Corey et al. [30]	-	58%	90%	83%	73%	0.830
**Kulkarni Model**						
Source	Cutoff	Sensitivity	Specificity	PPV	NPV	AUROC
Kulkarni et al. [31]	6.13	83.3%	94.6%	-	-	0.945
**ADAPT Score**						
Source	Cutoff	Sensitivity	Specificity	PPV	NPV	AUROC
Nielsen et al. [33]	>6.15	64.0%	75.0%	71.0%	68.0%	0.760
**MEFIB Index**						
Source	Cutoff	Sensitivity	Specificity	PPV	NPV	AUROC
Jung et al. [37]	MRE ≥ 3.3 kPa and FIB-4 index ≥ 1.6	50.0%	99.4%	83.2%	83.2%	0.900
**FAST Score**						
Source	Cutoff	Sensitivity	Specificity	PPV	NPV	AUROC
Woreta et al. [38]	0.35	91.0%	50.0%	51.0%	90.0%	0.807
	0.67	52.0%	87.0%	69.0%	76.0%	
	0.38	90.0%	53.0%	52.0%	90.0%	
	0.72	44.0%	90.0%	72.0%	73.0%	

Abbreviations: NAFLD—nonalcoholic fatty liver disease; PPV—positive predictive value; NPV—negative predictive value; AUROC—area under the receiver operating characteristic curve; ADAPT score—PRO-C3, presence of type 2 diabetes, platelet count, and age; PRO-C3—procollagen type-III N-terminal peptide; NFS—NAFLD fibrosis score; Kulkarni model—BMI, GGT, 25-OH-vitamin D and platelet count; MEFIB index—MRE in combination with FIB-4 index; FAST score—liver stiffness measurement and controlled attenuation parameter measured by VCTE (FibroScan) and serum levels of aspartate transaminase; MRE—magnetic resonance elastography; FIB-4 index—Fibrosis-4 index; VCTE—vibration-controlled transient elastography; general clinical features -age, BMI, sex, and diabetes status; GGT—gamma-glutamyl transferase levels.

**Table 9 diagnostics-12-02608-t009:** Diagnostic performance of novel biomarkers in the detection of advanced fibrosis.

**Cytokeratine-18 Fragment M30**						
Source	Cutoff	Sensitivity	Specificity	PPV	NPV	AUROC
Harrison et al. [20]	260 U/L	94%	23%	55%	80%	0.590
**Cytokeratine-18 Fragment M65**						
Source	Cutoff	Sensitivity	Specificity	PPV	NPV	AUROC
Harrison et al. [20]	545 U/L	95%	25%	56%	83%	0.600
**Procollagen Type-III N-Terminal Peptide**						
Source	Cutoff	Sensitivity	Specificity	PPV	NPV	AUROC
Bril et al. [28]	20 ng/mL	50%	96%	67%	92%	0.900
	13.2 ng/mL	88%	80%	43%	97%	
Nielsen et al. [33]	13.45 ng/mL	77%	59%	44%	87%	0.730
**Monocyte Chemoattractant Protein 1**						
Source	Cutoff	Sensitivity	Specificity	PPV	NPV	AUROC
Harrison et al. [20]	245.1	93%	14%	52%	67%	0.510
**Cohort-Specific Model** (**Serum CK-18, Fasting Insulin, Platelets Count, Sex, HbA1c**)						
Source	Cutoff	Sensitivity	Specificity	PPV	NPV	AUROC
Bril et al. [28]	<−2.613 and >1.015	88%	86%	57%	97%	0.860
	−1.369	80%	83%	45%	96%	
**Prognostic Factor Model**						
Source	Cutoff	Sensitivity	Specificity	PPV	NPV	AUROC
Caussy et al. [29]	-	90%	37%	20%	95%	0.840
**Top 10 Metabolite Panel**						
Source	Cutoff	Sensitivity	Specificity	PPV	NPV	AUROC
Caussy et al. [29]	-	90%	79%	43%	98%	0.940
**ADAPT Score**						
Source	Cutoff	Sensitivity	Specificity	PPV	NPV	AUROC
Nielsen et al. [33]	>6.16	78%	69%	50%	88%	0.800
**Combination of 6 Biomarkers**						
Source	Cutoff	Sensitivity	Specificity	PPV	NPV	AUROC
Bril et al. [28]	PRO-C3—13.2 ng/mL APRI—0.423 AST—38 units/L FIB-4 index—1.666 FibroTest—0.353 NFS −0.053	71%	94%	68%	95%	0.910

Abbreviations: PPV—positive predictive value; NPV—negative predictive value; AUROC—area under the receiver operating characteristic curve; CK-18—cytokeratine-18; prognostic factor model—score includes platelet count, international normalized ratio, HbA1c, and alkaline phosphatase based on backward stepwise elimination of factors associated with advanced fibrosis; serum top 10 metabolites panel—5 alpha-androstan-3 beta monosulfate, pregnanediol-3-glucuronide, androsterone sulfate, epiandrosterone sulfate, palmitoleate, dehydroisoandrosterone sulfate, 5 alpha-androstan-3 beta disulfate, glycocholate, taurine, and fucose; ADAPT score—PRO-C3, presence of type 2 diabetes, platelet count, and age; combination of 6 biomarkers—PRO-C3, APRI, AST, FIB-4 index, FibroTest, and NFS; PRO-C3—procollagen type-III N-terminal peptide; APRI—AST to platelet ratio index; AST—aspartate transaminase; NFS—NAFLD fibrosis score; FIB-4 index—Fibrosis-4 index; NAFLD—nonalcoholic fatty liver disease.

## Data Availability

Not applicable.

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
