# Peer review of "Accuracy of Noninvasive Diagnostic Tests for the Detection of Significant and Advanced Fibrosis Stages in Nonalcoholic Fatty Liver Disease: A Systematic Literature Review of the US Studies"

_diagnostics, 2022, doi:10.3390/diagnostics12112608_

Round 1
Reviewer 1 Report
This review summarizes reports of noninvasive liver fibrosis biomarkers for NAFLD in US. The paper has a chance to be good, but I must tell you that it contains serious misunderstandings and needs to be corrected.
1) Fibrosis in NAFLD is rarely assessed with METAVIR and more often with the NASH CRN fibrosis staging system (PMID: 29222917). Please specify the index of fibrosis used for each study. 
2) For studies indicated in the table, please specify the number of cases used in the study.
3) You stated ‘Only US patients were considered to avoid heterogeneity of the pooled sample that may impact the evidence synthesis’. I understand that the United States is a diverse country with a diverse set of racial and ethnic groups. Please correct this explanation.
4) The importance of identifying F2 is because F2 fibrosis is prognostically relevant. Please clearly state this. Also, early therapeutic intervention by identifying F1 fibrosis may have a certain positive impact (PMID: 35040549). How about adding this to the discussion?
Reviewer 2 Report
Literature published between 2016 and 2022 and concern about non-invasive diagnostic tools in detecting significant or advanced (F2/F3) fibrosis among patients with NAFLD in the US healthcare context were collected for review. The diagnostic accuracy for fibrosis were compared across different modalities. Imaging techniques with the highest diagnostic accuracy in F2/F3 detection and differentiation were magnetic resonance elastography and vibration-controlled transient elastography. The most promising standard blood biomarkers were NAFLD fibrosis score and FIB-4. The novel approaches in liver fibrosis detection may be improved by combining imaging techniques and blood biomarkers. However, they concluded that non-invasive techniques that provide a sufficiently sensitive and reliable estimate of fibrosis changes are still missing.
Comments
1. Many non-invasive modalities for fibrosis measurement have been developed before 2016. Therefore, this review is missing many original data that were published before 2016.
2. The review included reports from the US only. Please explain the difference in fibrosis measurements between other areas and the US.
3. Only 20 papers were included that may have lost its representative. Some modality has only one report been reviewed. Without direct comparison between two tests, it will be difficult to determine which test is superior to the other by accuracy calculation.
4. Figure 1 is well-known and may be deleted.
5. Some single tests such as AST, ALT and GGT are liver inflammation markers. These categories can be deleted.
Round 2
Reviewer 1 Report
This manuscript has been revised in a considerably better direction. I think it is equivalent to acceptance.
Reviewer 2 Report
The investigators addressed most of the comments.
No further comments.